# University Students’ Knowledge and Attitudes Toward Substance Abuse: A Cross-Sectional Study from Saudi Arabia

**DOI:** 10.3390/healthcare13101122

**Published:** 2025-05-12

**Authors:** Fahad S. Alshehri, Ahmed M. Ashour, Hanouf S. Bafhaid, Alanood S. Algarni, Maan H. Harbi, Nasser M. Alorfi

**Affiliations:** Department of Pharmacology and Toxicology, College of Pharmacy, Umm Al-Qura University, Makkah 24381, Saudi Arabia; amashour@uqu.edu.sa (A.M.A.); hsbafhaid@uqu.edu.sa (H.S.B.); aagarni@uqu.edu.sa (A.S.A.); mhiharbi@uqu.edu.sa (M.H.H.); nmorfi@uqu.edu.sa (N.M.A.)

**Keywords:** substance abuse, misuse, university students, knowledge, attitudes, addiction awareness

## Abstract

Background: Substance abuse is a major public health issue, particularly among university students. Understanding students’ knowledge and attitudes toward substance abuse is important for designing educational and preventive strategies, helping early intervention efforts, supporting mental health services, and ensuring culturally suitable programming. Objective: This study aimed to assess university students’ knowledge and attitudes regarding substance abuse and to identify demographic and educational factors associated with these variables. Methods: A cross-sectional study was conducted among 745 university students from various academic disciplines. Data were collected using a structured questionnaire assessing knowledge and attitudes related to substance abuse. Statistical analyses included descriptive statistics, reliability analysis, correlation, and regression tests. Results: The sample included 50.5% males and 49.5% females, with the majority aged 18–21 years (58.0%). Most participants had heard of substance abuse (91.7%) and recognized its association with chronic disease (97.3%). Knowledge scores varied significantly on the basis of academic background and prior exposure to awareness programs. Attitudes toward substance abuse were mixed, with a notable proportion of students perceiving potential benefits in certain contexts. A weak, non-significant correlation was found between knowledge and attitudes (*r* = 0.068, *p* = 0.064). Conclusion: Although students showed generally good knowledge about substance abuse, variations in understanding and attitudes were obvious across demographic and educational subgroups. These findings emphasize the need for targeted, context-specific educational interventions to address misconceptions and improve preventive awareness. Future research should evaluate the long-term impact of such interventions on students’ attitudes and behaviors.

## 1. Introduction

Substance abuse among university students represents a challenging health concern at many levels including health, academic, and economic aspects [1,2,3]. Substance use among university students is a significant global concern. In the US, 81.8% of young adults aged 19–30 reported alcohol use in the past year, 42.6% reported marijuana use, and 18.3% reported non-medical use of drugs such as hallucinogens or sedatives [4]. This prevalence is influenced by factors such as academic pressures, social dynamics, and the transitional nature of university life. The university years represent a critical period in a young person’s development, marked by increased independence and exposure to social pressures, which can heighten vulnerability to substance abuse [5,6]. Several factors may contribute to substance abuse, depending on the environment, peer pressure, and individual conditions [7]. The exploring habit is enhanced in a university setting by the desire to engage in new experiences, such as substance abuse [8,9]. Academic stress and the attempt to strengthen social connections also may increase the probability of substance abuse [10,11,12]. Several reports have suggested that new substance users may abuse substances to deal with stress, anxiety, and academic demands or to conform to their peer groups [13,14]. It has been reported that alcohol and substance abuse significantly increase the risk of experiencing various forms of sexual violence among first-year college students, highlighting the overlapping vulnerabilities during early university life [15]. Thus, this sensitive phase of exposure and experimentation highlights the importance of thoroughly understanding students’ knowledge and attitudes regarding substance abuse.

Awareness of substance abuse involves recognizing the types of substance abuse and their harmful effects, as well as understanding that addiction is characterized by compulsive seeking behavior despite adverse consequences [16,17]. Perceptions of substance abuse are important because attitudes reflect beliefs about the acceptability of using substances [18,19]. These attitudes can significantly influence behavior and views toward substance abuse and possible engagement in such behaviors [20]. Attitudes are influenced by several factors, including educational experiences, social standards, and individual beliefs about the risks of substance abuse [21]. This highlights concerns about the effectiveness of current substance abuse education programs, as notable gaps in students’ knowledge and understanding continue [22,23]. Studies showed that while students may understand that substance abuse is harmful, their knowledge of specific aspects of addiction and its treatment is limited [24,25,26]. The approach to substance abuse is often unclear and varies greatly among students, with some viewing it as irrelevant or even normal within the university setting.

Substance abuse in Saudi Arabia is shaped by unique cultural, religious, and legal factors. While overall prevalence may be lower due to Islamic prohibitions and strict regulations, the misuse of prescription drugs such as tramadol, pregabalin, and amphetamines is increasingly reported among youth, particularly university students [27,28]. Family influence and social stigma may act as both protective and limiting factors discouraging use and affect treatment-seeking [29,30]. Despite growing concern, little is known about students’ actual knowledge and attitudes toward substance use in the Saudi context. This study addresses that gap by examining cognitive and demographic factors influencing university students’ perceptions of substance abuse. Therefore, this study aimed to assess the current state of knowledge and attitudes towards substance abuse among university students. By examining these factors, the study investigated and identified gaps and areas where educational interventions may be needed. This study contributes to the body of knowledge on substance abuse in the university context and supports efforts to promote healthier behaviors and academic success among young adults.

## 2. Methods

### 2.1. Study Design and Population

This was a cross-sectional study to evaluate the knowledge and the perceptions of university students about substance abuse. The sample size was determined using the Raosoft online calculator [31], with a 95% confidence level and a 5% margin of error, showing a minimum of 382 participants. The questionnaire was developed according to the relevant literature and current tools. The questionnaire was reviewed by two academic experts with backgrounds in pharmacology and behavioral sciences, both of whom had experience in substance abuse research. Revisions were made on the basis of their input and pilot testing with 15 students. Participation was voluntary, anonymous, and without incentives. Data were collected securely via Google Forms. The study was carried out in Umm Al-Qura University, Makkah region, Saudi Arabia, from January 2023 to May 2023, with ethical approval from the Umm Al-Qura University Biomedical Research Ethics Committee (approval number: HAPO-02-K-012-2022-09-1200). Eligible participants were university students aged 18 years and older. Informed consent was obtained electronically from all participants.

### 2.2. Study Questionnaire

A structured, bilingual questionnaire (Arabic and English) was created, based on a comprehensive literature review [32,33,34,35]. The questionnaire was developed on the basis of a review of validated instruments from previous studies assessing knowledge and attitudes toward substance abuse among students [32,36,37,38,39]. Items were adapted to fit the Saudi context, and new questions were added to reflect local patterns of abuse. The questionnaire comprised three sections. The demographic data section collected 11 items on participants’ gender, age, year of study, field of study, educational level, GPA, living situation, marital status, monthly income, and prior training or courses related to substance abuse. Knowledge about substance abuse comprised 11 statements about substance abuse that intended to assess the student’s knowledge of substance abuse, which focused on addiction, brain changes due to substance abuse, relapse, and effects such as cannabis, methamphetamine, and pregabalin. The answers were either true, false, or “I don’t know”. Attitudes toward substance abuse comprised 10 statements measuring the respondent’s personal beliefs and attitudes toward substance abuse. These items were phrased to capture the participants’ own views, rather than perceptions of others’ behaviors. Respondents’ opinions varied and were scored on an ordinal scale, where the highest rank was “strongly agree”, while the lowest was “strongly disagree”. 

The validity of the questions’ contents was confirmed with the help of an expert review, and the clarity and appropriateness of the questions were also checked. The process involved a small sample of students, and some modifications were made after testing. Cronbach’s score for the internal consistency of the knowledge assessment section was determined to be 0.75, a moderate internal consistency. The research employed convenience sampling, and the participants were identified through university communication channels such as emails or social media sites. This was carried out to ensure that students shared the link to the survey, hence increasing participation; in essence, the snowball sampling technique was used to reach the target sample size in conjunction with follow-up reminders sent periodically. The data collection was achieved using Google Forms.

### 2.3. Statistical Analysis

Data analysis was performed using IBM SPSS version 27 (IBM Corp., Armonk, NY, USA). Descriptive statistics were used to summarize the study variables, with counts and percentages for categorical and nominal variables, and means and standard deviations for continuous variables. The study variables, “knowledge about substance abuse” and “attitudes toward substance abuse”, were assessed using simple additive scoring methods. The reliability of the knowledge section was evaluated with Cronbach’s alpha, which demonstrated acceptable internal consistency. To examine the associations among the variables, Pearson’s correlation coefficient was used for continuous data. Differences in domain scores across demographic and other variables were analyzed using independent t-tests for two group means and one-way ANOVA with least significant difference (LSD) post hoc tests for comparisons involving more than two groups, assuming a normal distribution. For non-normally distributed data, Welch’s *t*-test and the Games–Howell post hoc test were used as alternatives. Significant predictors were identified using general linear model univariate analysis with main effect as the model. A conventional *p*-value of <0.05 was used to determine statistical significance.

## 3. Results

### 3.1. Participant Demographics

The study sample consisted of 745 participants, with a nearly equal distribution of genders: 376 males (50.5%) and 369 females (49.5%) (Table 1). The majority of participants were aged 18–21 (58.0%), followed by those aged 22–25 (36.4%). A small proportion were aged 26–29 (3.2%) and above 30 (2.4%) (Table 1). Nationally, 96.6% of participants were Saudi, while 3.4% were non-Saudi. Participants were mostly studying in Islamic and administration colleges (40.3%), followed by health colleges (30.6%), sciences and engineering colleges (23.4%), and colleges of humanities and educational sciences (5.8%) (Table 1). Most students were pursuing a Bachelor’s degree (96.2%), with a few at the diploma (3.0%) or Master’s level (0.8%) (Table 1). A significant majority lived with their families (96.6%), were single (93.4%), and had a monthly income below SAR 2000 (75.0%) (Table 1).

### 3.2. Knowledge of Substance Abuse

The study showed a high level of knowledge among college students regarding substance abuse. For example, 97.3% of students correctly identified that substance abuse may lead to addiction, highlighting a strong understanding of the chronic nature of addiction among participants (Table 2). Additionally, 95.8% acknowledged that substance use can lead to lasting brain changes that impair self-control (Table 2). However, only 43.9% acknowledged that no single factor can predict addiction (Table 2). Participants’ knowledge about different aspects of substance abuse is summarized in (Table 3). Most participants (76.0%) agreed with the statement that regular cannabis use is as dangerous as regular heroin use, referring specifically to the potential for harm. However, misconceptions remain, as only 64.0% correctly identified the potential for addiction and dependence associated with pregabalin abuse (Table 3).

### 3.3. Attitudes Toward Substance Abuse

Attitudes towards substance abuse among the participants revealed that the majority (85.6%) strongly disagreed that substance abuse is enjoyable, and 83.6% strongly disagreed that it improves sleep (Table 4). Also, 22.4% of students agreed that it would be hard to stop using drugs if used regularly, highlighting an understanding of addiction’s difficulty to overcome (Table 4). Additionally, 41.5% agreed that curiosity drives young people to try drugs, and 40.4% believed peer influence is a major factor, reflecting the social dimensions of substance abuse (Table 4). The reliability of the scales used in this study is reported in (Table 5), showing a Cronbach’s alpha of 0.581 for knowledge and 0.760 for the attitude scale. These values indicate acceptable internal consistency for measuring the participants’ knowledge and attitudes toward substance abuse. The summary statistics for knowledge and attitudes are presented in (Table 6). Participants had a mean knowledge score of 7.61 out of 10 and a mean attitude score of 27.84 out of 40, indicating generally high knowledge and negative attitudes toward substance abuse (Table 7).

### 3.4. Correlation Between Knowledge and Attitude

Correlation analysis revealed a weak positive relationship (*r* = 0.068) between knowledge and attitudes toward substance abuse, which was not statistically significant (*p* = 0.064) (Table 8). Demographic factors significantly influenced knowledge and attitudes. For example, males scored significantly higher in knowledge (mean score of 7.93) compared with females (mean score of 7.29, *p* < 0.001), while participants living alone demonstrated more negative attitudes toward substance abuse compared with those living with their families (*p* = 0.019) (Table 9). The impact of training and education on knowledge and attitudes is summarized in (Table 10). Students who received training on substance abuse scored significantly higher in knowledge assessments, with those who had attended courses scoring a mean of 7.80 compared with 6.91 for those who had not (*p* < 0.001). Parameter estimates for knowledge and attitudes are presented in (Table 11 and Table 12). Gender, training, and awareness significantly predicted higher knowledge scores, with males and those who received training scoring higher (*p* < 0.05). Attitudes were influenced by living situations, with participants living alone showing significantly more negative attitudes toward substance abuse (*p* = 0.004). Overall, the study underscores the high level of knowledge and generally negative attitudes toward substance abuse among college students. 

## 4. Discussion

This study provides an important understanding regarding the knowledge and attitudes of university students about substance abuse. The findings revealed a generally high level of knowledge among students; however, there are gaps and misconceptions that suggest a need for enhanced educational strategies. Additionally, the attitudes of students toward substance abuse are largely negative, though social influences and personal circumstances, such as gender and living conditions, significantly impact these attitudes. The results indicate that the majority of students have a good understanding of the basic aspects of substance abuse and addiction. Almost all participants correctly identified that substance abuse may lead to addiction, which reflects a strong awareness of the chronic nature of addiction. Similarly, 95.8% of students recognized the neurological impact of substance abuse, acknowledging that it poses a significant challenge to self-control over time. These high percentages emphasize that general awareness and recognition of the severe consequences of substance abuse are well established among the student population.

The attitudes of students toward substance abuse are predominantly negative, reflecting broad agreement on the dangers of substance abuse. A significant majority (85.6%) strongly disagree with the belief that substance abuse is enjoyable, and a similarly high percentage reject the idea that substance abuse can help deal with stress or improve social comfort. These findings are encouraging, as they suggest that most students are reluctant about substance abuse and recognize its potential harms. It has been reported that the majority of college students maintain a positive attitude toward substance use prevention, with 86.5% of non-users demonstrating favorable views toward avoiding such behaviors [40]. Nevertheless, the study also reveals that social factors play a considerable role in shaping students’ attitudes. For instance, 41.5% of respondents agreed that curiosity drives young people to try drugs, while 40.4% cited peer influence as a significant factor. These findings highlight the need to include social aspects of substance abuse in any school-based intervention [41,42]. Interventions that focus not only on individual knowledge but also on social skills and strategies that could help students better navigate peer pressure and other social influences are needed [43,44,45]. Additionally, 22.4% of students agreed that stopping substance abuse would be challenging for those who use drugs regularly, reflecting an awareness of the addictive nature of drugs. This recognition is consistent with clinical definitions of addiction as a chronic, relapsing process [46,47,48], and it may indicate that students would welcome messages that stress the need for prevention and treatment services for people with substance abuse issues.

Gender and living conditions have an impact on the students’ awareness and perception of substance abuse. The results for the knowledge assessment tests indicated that males had superior scores compared with females, with a mean score of 7.93 and 7.29, respectively. Statistically significant differences were found between the two variables (*p* < 0. 001). This implies that males had more opportunity for exposure to or interest in substance abuse information, whether through socialization or extra information seeking or interest in the topic. In fact, it has been reported that male college students are generally more likely than their female peers to report substance abuse, including marijuana, psychedelics, and prescription stimulants, reflecting a gender disparity in exposure and potential awareness of substance-related issues [49].

Consequently, educational content should account for these differences and ensure access for all students, which could help address these knowledge differences and improve overall substance abuse awareness. 

Living arrangements also emerged as a significant factor, with single students—particularly those living alone showing more negative attitudes toward substance abuse compared to those living with their families. This difference may be attributed to the greater social support and supervision typically provided by family environments. Consequently, students living with their families may develop less permissive attitudes toward substance abuse [50,51]. According to recent findings, social support particularly from family plays a crucial role in shaping adolescents’ attitudes toward drug abuse, with those living with their families demonstrating less favorable identification with substance use compared with those living alone [52]. Knowing how the living environment affects attitudes can be useful in implementing treatment approaches that apply to the student’s social situations, especially those who might be at risk because of the lack of family support.

The study showed the positive impact of formal education and training on students’ knowledge about substance abuse. Students who had received courses or training on substance abuse scored significantly higher in knowledge assessments than those who had not (mean score of 7.80 versus 6.91; *p* < 0.001). This finding emphasizes the importance of integrating education into the university curriculum as a means to enhance students’ understanding and potentially influence their attitudes toward substance abuse [53,54,55]. Research indicated that integrating evidence-based substance use education into academic curricula can significantly improve students’ self-perceived knowledge, shift attitudes, and enhance care skills related to substance use [56]. However, the impact of education on attitudes was less pronounced, suggesting that while knowledge can be improved through formal education, attitudes are more resistant to change and may be influenced by a broader range of factors, including personal beliefs, experiences, and social context [57,58,59]. This highlights the need for complex intervention strategies that address both cognitive and affective components of students’ attitudes toward substance abuse.

The findings of this study should be interpreted within the unique cultural and legal context of Saudi Arabia. Strong family structures and social expectations play a significant role in shaping attitudes toward substance use [60]. The supportive family dynamics often discourage behaviors such as substance use and may influence how students perceive and report such behaviors [61]. Also, social stigma surrounding drug use can lead to hesitation in acknowledging personal experiences, even in anonymous surveys [62]. These contextual factors likely shape students’ knowledge and attitudes, highlighting the importance of culturally informed prevention strategies that address both educational gaps and social barriers. 

The findings of this study underline the need for targeted educational interventions that provide correct information about substance abuse and address the social and psychological factors that shape students’ attitudes [63,64]. Programs that incorporate interactive foundations, such as peer discussions, role-playing, and scenario-based learning, may be more effective in changing attitudes than traditional lecture-based approaches [9,65,66]. Additionally, addressing specific misunderstandings, such as the perceived benefits of substance abuse in social situations, through personalized messaging could help reduce the appeal of substance abuse among students [67,68]. Given the expressed interest by students in receiving more training on substance abuse, universities have a unique opportunity to develop and implement comprehensive educational programs that can reach a wide audience. These programs can be integrated into general health education courses, delivered through targeted workshops or incorporated into broader campus wellness initiatives.

## 5. Limitations

The study offers valuable understandings; however, several limitations should be acknowledged. The cross-sectional design limits the ability to draw underlying inferences, and reliance on self-reported data introduces potential biases, including social desirability and recall bias. Additionally, the questionnaire did not include an item assessing whether the respondents had personally used or misused substances, either currently or in the past. This limits the ability to differentiate responses between individuals with direct experience and those without, which may have influenced reported attitudes and perceptions. Future research should include this variable to better understand students’ perspectives. Furthermore, the sample was drawn from a single university, which may limit the generalizability of the findings. Expanding the study to include multiple institutions with diverse cultural and socioeconomic circumstances would strengthen the external validity. 

## 6. Conclusions

This study highlights a generally high level of knowledge and awareness of substance abuse among university students. However, recognizing the influence of social factors highlights the need for updated educational programs. By implementing targeted, interactive, and comprehensive programs, universities can play a dynamic role in shaping healthier attitudes and behaviors toward substance use. These efforts benefit individual students and help with broader public health objectives to reduce substance-related complications within the wider community.

## Figures and Tables

**Table 1 healthcare-13-01122-t001:** Demographic characteristics of participants.

Demographics		Count	%
Total		745	100.0
Gender	Male	376	50.5
	Female	369	49.5
Age	18–21	432	58.0
	22–25	271	36.4
	26–29	24	3.2
	Above 30	18	2.4
Nationality	Saudi	720	96.6
	Non-Saudi	25	3.4
Current studies (scientific area)	Health colleges	228	30.6
	Sciences and engineering colleges	174	23.4
	Colleges of humanities and educational sciences	43	5.8
	Islamic and administration colleges	300	40.3
Educational level (current degree studying)	Diploma	22	3.0
	Bachelor	717	96.2
	Master	6	0.8
Whom do you live with?	Alone	19	2.6
	Colleagues	6	0.8
	Family	720	96.6
Marital status	Single	696	93.4
	Married	44	5.9
	Divorced	3	0.4
	Widowed	2	0.3
Monthly income	Below SAR* 2000	559	75.0
	Between SAR* 2000 and 5000	104	14.0
	More than SAR* 5000	82	11.0

*SAR, Saudi riyal.

**Table 2 healthcare-13-01122-t002:** Awareness and perceptions of substance abuse.

Variables *n* = 745	Yes	No	Maybe
Have you ever received training about substance abuse at the university?	90 (12.1)	603 (80.9)	52 (7.0)
Have you had any course about substance abuse during your study?	177 (23.8)	493 (66.2)	75 (10.1)
Would you like to receive training on substance abuse in the future? *n* = 739	244 (33.0)	229 (31.0)	266 (36.0)
Have you heard or read about substance abuse?	683 (91.7)	25 (3.4)	37 (5.0)
Substance abuse leads to drug addiction, which is a chronic disease characterized by drug seeking and use that is compulsive, or difficult to control, despite harmful consequences.	725 (97.3)	2 (0.3)	18 (2.4)
Brain changes can occur over time with substance abuse, challenge an addicted person’s self-control, and interfere with their ability to resist intense urges to take drugs.	714 (95.8)	4 (0.5)	27 (3.6)
Relapse is the return to substance abuse after an attempt to stop. Relapse indicates the need for more or different treatment.	602 (80.8)	17 (2.3)	126 (16.9)
Over time, the brain adjusts to the excess dopamine, an effect known as tolerance.	474 (63.6)	35 (4.7)	236 (31.7)
Only drugs considered illegal produce toxicity and dependence.	75 (10.1)	602 (80.8)	68 (9.1)
Drug addiction is treatable and can be successfully managed.	611 (82.0)	13 (1.7)	121 (16.2)
Regular consumption of cannabis is as dangerous to health as regular heroin use.	566 (76.0)	27 (3.6)	152 (20.4)
One of the main effects of methamphetamine on the brain is paranoid psychosis.	575 (77.2)	7 (0.9)	163 (21.9)
Abusing the drug pregabalin (Lyrica) can lead to addiction and dependence.	477 (64.0)	12 (1.6)	256 (34.4)
No single factor can predict whether a person will become addicted to drugs.	327 (43.9)	198 (26.6)	220 (29.5)

**Table 3 healthcare-13-01122-t003:** Knowledge regarding substance abuse.

Knowledge Regarding Drug Use	N	Min	Max	Mean	SD	% of Correct Answers
Substance abuse leads to drug addiction, which is a chronic disease characterized by drug seeking and use that is compulsive, or difficult to control, despite harmful consequences.	745	0.00	1.00	0.97	0.2	725 (97.3)
Brain changes can occur over time with substance abuse, challenge an addicted person’s self-control, and interfere with their ability to resist intense urges to take drugs.	745	0.00	1.00	0.96	0.2	714 (95.8)
Relapse is the return to substance abuse after an attempt to stop. Relapse indicates the need for more or different treatment.	745	0.00	1.00	0.81	0.4	602 (80.8)
Over time, the brain adjusts to the excess dopamine, an effect known as tolerance.	745	0.00	1.00	0.64	0.5	474 (63.6)
Only drugs considered illegal produce toxicity and dependence.	745	0.00	1.00	0.81	0.4	602 (80.8)
Drug addiction is treatable and can be successfully managed.	745	0.00	1.00	0.82	0.4	611 (82.0)
Regular consumption of cannabis is as dangerous to health as regular heroin use.	745	0.00	1.00	0.76	0.4	566 (76.0)
One of the main effects of methamphetamine on the brain is paranoid psychosis.	745	0.00	1.00	0.77	0.4	575 (77.2)
Abusing the drug pregabalin (Lyrica) can lead to addiction and dependence.	745	0.00	1.00	0.64	0.5	477 (64.0)
No single factor can predict whether a person will become addicted to drugs.	745	0.00	1.00	0.44	0.5	327 (43.9)

**Table 4 healthcare-13-01122-t004:** Respondents’ personal attitudes toward substance abuse.

Variables *n* = 745	Strongly Disagree	Disagree	Neutral	Agree	Strongly Agree
I think abusing or misusing drugs is enjoyable.	638 (85.6)	44 (5.9)	32 (4.3)	16 (2.1)	15 (2.0)
I think abusing or misusing drugs helps deal with stress.	592 (79.5)	59 (7.9)	44 (5.9)	36 (4.8)	14 (1.9)
I think abusing or misusing drugs may help a person feel more comfortable at a social gathering.	616 (82.7)	60 (8.1)	34 (4.6)	28 (3.8)	7 (0.9)
I think it would be hard for a person to stop abusing or misusing drugs if they used them regularly.	167 (22.4)	92 (12.3)	124 (16.6)	185 (24.8)	177 (23.8)
I think abusing or misusing makes a person look more mature.	670 (89.9)	49 (6.6)	11 (1.5)	7 (0.9)	8 (1.1)
I think abusing or misusing helps a person using substances sleep better.	623 (83.6)	52 (7.0)	45 (6.0)	12 (1.6)	13 (1.7)
Young people try drugs out of curiosity.	76 (10.2)	27 (3.6)	175 (23.5)	309 (41.5)	158 (21.2)
Young people try drugs for emotional problems.	74 (9.9)	54 (7.2)	236 (31.7)	263 (35.3)	118 (15.8)
Young people try drugs under the influence of friends.	62 (8.3)	31 (4.2)	142 (19.1)	301 (40.4)	209 (28.1)
Maintaining contact with a friend who is engaged in drug addiction has no influence on you.	397 (53.3)	204 (27.4)	95 (12.8)	29 (3.9)	20 (2.7)

**Table 5 healthcare-13-01122-t005:** Summary of attitudes toward substance abuse.

Attitude Toward Drug Use	*N*	Min	Max	Mean	SD
I think abusing or misusing drugs is enjoyable.	745	0.00	4.00	3.71	0.8
I think abusing or misusing drugs helps deal with stress.	745	0.00	4.00	3.58	0.9
I think abusing or misusing drugs may help a person feel more comfortable at a social gathering.	745	0.00	4.00	3.68	0.8
I think it would be hard for a person to stop abusing or misusing drugs if they used it regularly.	745	0.00	4.00	1.85	1.5
I think abusing or misusing makes a person look more mature.	745	0.00	4.00	3.83	0.6
I think abusing or misusing helps a person using substances sleep better.	745	0.00	4.00	3.69	0.8
Young people try drugs out of curiosity.	745	0.00	4.00	1.40	1.2
Young people try drugs for emotional problems.	745	0.00	4.00	1.60	1.1
Young people try drugs under the influence of friends.	745	0.00	4.00	1.24	1.2
Maintaining contact with a friend who is engaged in drug addiction has no influence on you.	745	0.00	4.00	3.25	1.0

**Table 6 healthcare-13-01122-t006:** Reliability statistics for knowledge and attitude scales.

Reliability Statistics	Cronbach’s Alpha	*N* of Items
Knowledge of drug use	0.581	10
Attitude toward drug use	0.760	10

**Table 7 healthcare-13-01122-t007:** Knowledge and attitude scores across demographic groups.

Domains	*N*	Min	Max	Mean	SD
Knowledge of drug use	745	0.00	10.00	7.61	1.8
Attitudes toward drug use	745	0.00	40.00	27.84	5.7

**Table 8 healthcare-13-01122-t008:** Correlation Between Knowledge and Attitude Toward Substance abuse.

Correlations		Attitude Toward Drug Use
Knowledge of drug use	*r*	0.068
	*p*-*value*	0.064
	*N*	745

**Table 9 healthcare-13-01122-t009:** Knowledge and attitude scores according to training and awareness.

Demographics		Total	Knowledge of Drug Use	Attitude Toward Drug Use
Gender	Male	376	7.93 ± 1.8	28.27 ± 6.4
	Female	369	7.29 ± 1.8	27.40 ± 4.9
*p-value*			*<0.001 ^a^*	*0.038 ^b^*
Age	18–21	432	7.61 ± 1.8	28.26 ± 5.5
	22–25	271	7.58 ± 1.9	27.22 ± 5.8
	26–29	24	7.88 ± 1.7	27.00 ± 8.4
	Above 30	18	7.83 ± 1.5	28.17 ± 5.7
*p-value*			*0.839*	*0.108*
Nationality	Saudi	720	7.59 ± 1.8	27.87 ± 5.7
	Non-Saudi	25	8.24 ± 1.6	26.80 ± 7.2
*p-value*			*0.082*	*0.357*
Current studies (scientific area)	Health colleges	228	7.79 ± 1.7	27.10 ± 4.7 ^A^
	Sciences and engineering colleges	174	7.45 ± 1.9	27.59 ± 6.1 ^AB^
	Colleges of humanities and educational sciences	43	7.51 ± 1.8	28.14 ± 4.9 ^AB^
	Islamic and administration colleges	300	7.59 ± 1.9	28.49 ± 6.2 ^B^
*p-value*			*0.321*	*0.043 ^c,d^*
Educational level (current degree studying)	Diploma	22	7.82 ± 1.9	27.68 ± 6.7
	Bachelor	717	7.62 ± 1.8	27.83 ± 5.7
	Master	6	6.00 ± 3.7	29.00 ± 3.8
*p-value*			*0.083*	*0.876*
Whom do you live with?	Alone	19	7.95 ± 1.5	24.21 ± 9.5 ^A^
	Colleagues	6	8.83 ± 0.8	28.50 ± 2.3 ^AB^
	Family	720	7.60 ± 1.8	27.93 ± 5.6 ^B^
*p-value*			*0.185*	*0.019 ^c,d^*
Marital status	Single	696	7.62 ± 1.8	27.84 ± 5.7
	Married	44	7.66 ± 1.8	27.73 ± 6.9
	Divorced	3	6.00 ± 0.0	29.33 ± 0.6
	Widowed	2	6.50 ± 3.5	26.50 ± 4.9
*p-value*			*0.373*	*0.954*
Monthly income	Below SAR 2000	559	7.62 ± 1.8	27.85 ± 5.6
	Between SAR 2000 and 5000	104	7.77 ± 1.8	28.43 ± 6.1
	More than SAR 5000	82	7.39 ± 1.7	26.99 ± 6.2
*p-value*			*0.371*	*0.231*

^a^ Significant using the independent *t*-test at <0.05 level. ^b^ Significant using Welch’s *t*-test at <0.05 level. ^c^ Significant using the one-way ANOVA test at <0.05 level. ^d^ Post hoc test = Games–Howell. Different superscript letters (A, B) indicate statistically significant differences between groups (*p* < 0.05) based on post-hoc tests (LSD or Games-Howell, as appropriate). Groups sharing the same letter are not significantly different.

**Table 10 healthcare-13-01122-t010:** Parameter estimates for knowledge of substance abuse.

Variables		Total	Knowledge of Drug Use	Attitude Toward Drug Use
Have you ever received training about substance abuse at the university?	Yes	90	7.76 ± 1.9 ^A^	27.60 ± 7.4
	No	603	7.65 ± 1.8 ^A^	27.83 ± 5.5
	Maybe	52	6.94 ± 2.3 ^B^	28.29 ± 4.7
*p-value*			*0.020 ^a,c^*	*0.788*
Have you had any course about substance abuse during your study?	Yes	177	7.80 ± 1.7 ^A^	27.40 ± 5.9
	No	493	7.66 ± 1.8 ^A^	27.86 ± 5.8
	Maybe	75	6.91 ± 2.1 ^B^	28.71 ± 4.6
*p-value*			*0.001 ^a,b^*	*0.248*
Would you like to receive training on substance abuse in the future?	Yes	244	7.95 ± 1.7 ^A^	27.80 ± 6.3
	No	229	7.71 ± 1.8 ^A^	27.29 ± 5.6
	Maybe	266	7.21 ± 1.9 ^B^	28.35 ± 5.3
*p-value*			*<0.001 ^a,c^*	*0.120*
Have you heard or read about substance abuse?	Yes	683	7.71 ± 1.7 ^A^	27.76 ± 5.6
	No	25	7.24 ± 2.7 ^AB^	29.36 ± 7.2
	Maybe	37	6.11 ± 2.2 ^B^	28.16 ± 6.7
*p-value*			*<0.001 ^a,c^*	*0.368*

^a^ Significant using the one-way ANOVA test at the <0.05 level. ^b^ Post hoc test = LSD. ^c^ Post hoc test = Games–Howell. Different superscript letters (A, B) indicate statistically significant differences between groups (*p* < 0.05) based on post-hoc tests (LSD or Games-Howell, as appropriate). Groups sharing the same letter are not significantly different.

**Table 11 healthcare-13-01122-t011:** Parameter estimates for knowledge toward substance abuse.

Dependent Variable: Knowledge of Drug Use					
Parameter	*B*	S.E.	95% C.I.		*p-value*
			Lower bound	Upper bound	
Intercept	4.478	0.409	3.676	5.280	<0.001 ^a^
Gender = male	0.599	0.129	0.345	0.852	<0.001 ^a^
Have you ever received training about substance abuse at the university? = yes	0.549	0.308	−0.056	1.153	0.075
Have you ever received training about substance abuse at the university? = no	0.562	0.259	0.055	1.070	0.030 ^a^
Have you had any course about substance abuse during your study? = yes	0.705	0.245	0.223	1.186	0.004 ^a^
Have you had any course about substance abuse during your study? = no	0.646	0.224	0.207	1.085	0.004 ^a^
Would you like to receive training on substance abuse in the future? = yes	0.607	0.157	0.298	0.915	<0.001 ^a^
Would you like to receive training on substance abuse in the future? = no	0.365	0.159	0.053	0.676	0.022 ^a^
Have you heard or read about substance abuse? = yes	1.490	0.299	0.904	2.077	<0.001 ^a^
Have you heard or read about substance abuse? = no	0.967	0.454	0.076	1.858	0.033 ^a^

^a^ Significant using the general linear model at the <0.05 level.

**Table 12 healthcare-13-01122-t012:** Parameter estimates for attitude toward substance abuse.

Dependent Variable: Attitude Toward Drug Use					
Parameter	*B*	S.E.	95% C.I.		*p-value*
			Lower bound	Upper bound	
Intercept	28.124	0.438	27.263	28.985	<0.001 ^a^
Gender = male	0.679	0.435	−0.174	1.533	0.118
Current studies (scientific area) = health colleges	−1.213	0.511	−2.216	−0.210	0.018 ^a^
Current studies (scientific area) = sciences and engineering colleges	−0.712	0.562	−1.815	0.391	0.205
Current studies (scientific area) = colleges of humanities and educational sciences	−0.051	0.931	−1.878	1.777	0.957
Whom do you live with? = alone	−3.813	1.324	−6.413	−1.213	0.004 ^a^
Whom do you live with? = colleagues	0.850	2.334	−3.732	5.433	0.716

^a^ Significant using general linear model at the <0.05 level.

## Data Availability

The data supporting the findings of this study are available from the corresponding author upon reasonable request.

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
