# Peer review of "University Students’ Knowledge and Attitudes Toward Substance Abuse: A Cross-Sectional Study from Saudi Arabia"

_healthcare, 2025, doi:10.3390/healthcare13101122_

Round 1

Reviewer 1 Report

Comments and Suggestions for Authors

Dear authors,

Many thanks for submitting this paper on your esearch.

This topic is of interest and relevance and adds to the evoidence base looking at student exposure and thoughts/beliefs on substance use.

A few comments and thoughts:

The aim was clear and well presented.

Need to have consistency in the text, "Substance abuse" and "substance abuse" have been used through the text, it does not require capitals, please review and amend. Similarly Methamphetamine does not require a capital. Please check through for the use of capitals. Pregabalin is also incorrectly spelt

Page 1 Introduction (3rd last line) - "beginers" may be better repharsed to "new substance users"

Page 3 Results - (2nd paragraph) substance abuse does not neccessarily "lead to addiction" but may lead to addiction

Page 3 Results - (2nd paragraph) Can you quantify "dangerous" in respect to? Is is harm or is it dependence?

Page 3 Results - (3rd paragraph) looking at the question on the Table (Table 4) the wording was "makes me sleep better" this is very personable, should this be "helps a person using substances sleep better"

Results Table 4 - again this is not clear: "I think abusing or misusing subtances is enjoyable." is this the respondee or a person using subtances?

Results - from the results and information, the respondee is not asked if they have personally misused/abused a substance (either currently on in the past) I think this is a question that should be asked as thie response to this may affect the responses given as they will have a different perspective to respondees who have never used a substance. This could lead to a very different set of results for those who have and have not used a substance before. (this could be a limitation of the paper)

I think there should be section headers for Limitations/strengths and Conclusion.

The paper is quite heavy with stats which can make it hard to read and follow in some sections and this was supported by a large number of tables.

Overall, a good paper with good information but needs to be a bit clearer with regards to  certain points and a final grammar check.

I think the paper adds to the evidence base and provided added insight. I think it merits publication after a few amendents.

Best wishes

Author Response

Reviewer1

Dear authors,

Many thanks for submitting this paper on your research.

This topic is of interest and relevance and adds to the evidence base looking at student exposure and thoughts/beliefs on substance use.

A few comments and thoughts:

  1. The aim was clear and well presented.

We thank the reviewer.

  1. Need to have consistency in the text, "Substance abuse" and "substance abuse" have been used through the text, it does not require capitals, please review and amend. Similarly Methamphetamine does not require a capital. Please check through for the use of capitals. Pregabalin is also incorrectly spelt

We thank the reviewer for this important observation. We have carefully reviewed the entire manuscript to ensure consistency in the use of lowercase for terms such as “substance abuse” and corrected all improper capitalizations. Additionally, the spelling of "pregabalin" has been corrected throughout.

  1. Page 1 Introduction (3rd last line) - "beginers" may be better repharsed to "new substance users"

The word “beginers” has been corrected and replaced with “new substance users” as suggested to improve clarity and accuracy.

  1. Page 3 Results - (2nd paragraph) substance abuse does not neccessarily "lead to addiction" but may lead to addiction

We agree with the reviewer and have revised the sentence to state that “substance abuse may lead to addiction,” which more accurately reflects the risk without implying certainty.

  1. Page 3 Results - (2nd paragraph) Can you quantify "dangerous" in respect to? Is is harm or is it dependence?

We appreciate the reviewer’s request for clarification. The original statement was intended to refer specifically to potential for harm, not dependence. We have revised the text to clearly reflect this.

  1. Page 3 Results - (3rd paragraph) looking at the question on the Table (Table 4) the wording was "makes me sleep better" this is very personable, should this be "helps a person using substances sleep better"

Thank you for this suggestion. We have clarified the item wording in the results narrative to reflect that it was a personal perception. This has been addressed and noted in the manuscript.

  1. Results Table 4 - again this is not clear: "I think abusing or misusing subtances is enjoyable." is this the respondee or a person using subtances?

We appreciate the reviewer’s observation. The items in the attitude section, including "I think abusing or misusing substances is enjoyable," were intended to reflect the respondent’s own beliefs or perceptions. To clarify this, we have revised the Methods section to explicitly state that these statements capture personal attitudes. Additionally, we have updated the Table 4 caption to indicate that the responses represent the participants’ own views.

  1. Results - from the results and information, the respondee is not asked if they have personally misused/abused a substance (either currently on in the past) I think this is a question that should be asked as thie response to this may affect the responses given as they will have a different perspective to respondees who have never used a substance. This could lead to a very different set of results for those who have and have not used a substance before. (this could be a limitation of the paper)

We thank the reviewer for this insightful comment. We agree that the personal history of substance use could meaningfully influence participants’ attitudes and perceptions. Unfortunately, our questionnaire did not include a direct question regarding participants’ own history of substance use, which limits our ability to differentiate between users and non-users. This limitation has now been acknowledged explicitly in the revised manuscript under the newly added "Limitations and Strengths" section, and we have recommended that future studies incorporate such measures to better contextualize students' responses.

  1. I think there should be section headers for Limitations/strengths and Conclusion.

We thank the reviewer for this helpful suggestion. We have now added clear section headers titled “Limitations” and “Conclusion” to improve the structure and readability of the manuscript. These additions help guide the reader through the discussion and reinforce the study’s key findings and implications.

  1. The paper is quite heavy with stats which can make it hard to read and follow in some sections and this was supported by a large number of tables.

We appreciate the reviewer’s observation. We acknowledge that the manuscript contains substantial statistical data due to the nature of the study and the breadth of variables analyzed. However, we believe the level of detail is important to ensure transparency, reproducibility, and robustness of the findings. The tables were included to provide comprehensive data for readers who wish to examine the results in depth, while the narrative focuses on summarizing key findings to maintain readability. We hope this balance is acceptable to the journal and readers

  1. Overall, a good paper with good information but needs to be a bit clearer with regards to certain points and a final grammar check.

We thank the reviewer for the positive feedback and constructive remarks. In response, we have carefully revised the manuscript for improved clarity, coherence, and consistency. A thorough grammar and language check has also been performed to enhance readability and ensure the manuscript meets the journal’s linguistic standards.

  1. I think the paper adds to the evidence base and provided added insight. I think it merits publication after a few amendents.

We sincerely thank the reviewer for the encouraging comments and recognition of our study's contribution. We have carefully addressed all suggested amendments to improve the manuscript’s clarity, accuracy, and presentation, and we appreciate your support toward its potential publication.

Best wishes

Reviewer 2 Report

Comments and Suggestions for Authors

The authors of the manuscript “University Students’ Knowledge and Attitudes Toward Substance Abuse: A Cross-Sectional Study from Saudi Arabia” (healthcare-3589279) presented a cross-sectional study analysing university students’ knowledge and attitudes regarding substance abuse and identifying demographic and educational factors associated with them.

The authors presented a vital piece of research on the subject. However, a few issues are present:

Introduction:

Please add more specific information about the context of substance abuse in Saudi Arabia. Are there any particular substances that are more often used than in other parts of the world? Are there specific cultural factors that influence substance use and abuse? Additionally, clearly state the knowledge gap this study fills. What remains unknown about students' knowledge, attitudes, and associated factors regarding SA in Saudi Arabia? Broaden the literature review. Cite studies from diverse geographical contexts and also include region-specific studies if available.

Methods:

Please cite the “online calculator” used to determine sample size. Describe the “help of an expert” and “appropriateness” of questions, and how exactly the questions were formed? Were students ensured anonymity?  Were there any incentives?

Results:

Provide descriptive statistics for all demographic variables in Table 1. Include tables in the results; placing them at the end lowers their readability. Add footnotes to the tables.

Discussion:

Expand on the cultural and contextual relevance of findings, especially given the Saudi setting (e.g., religious norms, family structures, legal restrictions on drugs).

Consider limitations of self-reported data, especially in conservative settings where honesty may be affected.

Overall, this is an interesting study with potential for usage in an educational context. However, in the present format, it meets the parameters of a brief communication, not a research article. The discussion and introduction must be rewritten to include more comprehensive contextual information and an in-depth discussion of the findings.

Author Response

Reviewer 2

The authors of the manuscript “University Students’ Knowledge and Attitudes Toward Substance Abuse: A Cross-Sectional Study from Saudi Arabia” (healthcare-3589279) presented a cross-sectional study analysing university students’ knowledge and attitudes regarding substance abuse and identifying demographic and educational factors associated with them.

The authors presented a vital piece of research on the subject. However, a few issues are present:

Introduction:

  1. Please add more specific information about the context of substance abuse in Saudi Arabia. Are there any particular substances that are more often used than in other parts of the world? Are there specific cultural factors that influence substance use and abuse? Additionally, clearly state the knowledge gap this study fills. What remains unknown about students' knowledge, attitudes, and associated factors regarding SA in Saudi Arabia? Broaden the literature review. Cite studies from diverse geographical contexts and also include region-specific studies if available.

We appreciate the reviewer’s insightful suggestions. We have revised the Introduction to include a more detailed overview of the substance abuse landscape in Saudi Arabia.

Methods:

  1. Please cite the “online calculator” used to determine sample size. Describe the “help of an expert” and “appropriateness” of questions, and how exactly the questions were formed? Were students ensured anonymity? Were there any incentives?

We thank the reviewer for these valuable points. All requested details regarding sample size calculation, questionnaire development, expert review, anonymity, and incentives have been addressed and clarified in the revised Methods section.

Results:

  1. Provide descriptive statistics for all demographic variables in Table 1. Include tables in the results; placing them at the end lowers their readability. Add footnotes to the tables.

We thank the reviewer for this helpful suggestion. Table 1 has been reviewed and contains complete descriptive statistics for all demographic variables. A clarifying footnote has been added.

Discussion:

  1. Expand on the cultural and contextual relevance of findings, especially given the Saudi setting (e.g., religious norms, family structures, legal restrictions on drugs).

We thank the reviewer for this valuable suggestion. We have expanded the Discussion section to include a focused paragraph on the cultural and legal context in Saudi Arabia.

  1. Consider limitations of self-reported data, especially in conservative settings where honesty may be affected.

We thank the reviewer for this valuable point. We have expanded the Limitations section.

  1. Overall, this is an interesting study with potential for usage in an educational context. However, in the present format, it meets the parameters of a brief communication, not a research article. The discussion and introduction must be rewritten to include more comprehensive contextual information and an in-depth discussion of the findings.

We sincerely thank the reviewer for the constructive feedback. We have significantly revised both the Introduction and Discussion sections. The Introduction now provides more comprehensive contextual information, including local substance use trends, cultural factors, and a clearer articulation of the knowledge gap this study addresses. The Discussion section has been expanded to include deeper interpretation of the findings and their relevance within the Saudi cultural and legal context. We believe these revisions enhance the manuscript’s depth and align it with the expectations of a full research article.

Reviewer 3 Report

Comments and Suggestions for Authors

The study addresses a topic of global relevance to public health, as it deals with the knowledge and attitudes of university students towards illicit substance abuse, which has serious consequences for physical, social and mental well-being. The abstract provides a clear summary of the context, methods, results and conclusions of the study. It is concise and covers the essential elements of the research.
The introduction makes reference to the literature, the objectives of the study are clearly defined. However, a more explicit statement of the importance of the study and how it contributes to existing knowledge would be useful. In addition, it may also be useful to expand on the global context of knowledge and attitudes related to substance abuse among university students, namely statistical data, and how these trends may be relevant to Saudi Arabia. 

The study methodology has several strengths, including a well-described sample size and study design. In addition, ethical considerations, including approval and consent procedures, are adequately addressed. Although the period of the study is mentioned, it would be beneficial to discuss why this period was chosen and how it may impact knowledge and attitudes about substance abuse among university students.
The results of the study were presented clearly, with detailed demographic information, easy-to-read and interpret tables and statistical tests appropriate to the aims of the study. However, it may be useful to organize the results section more cohesively, possibly using subheadings to improve readability.

The discussion section effectively relates the results of the study to the literature, referencing other studies on knowledge and attitudes about substance abuse in university students. However, they may reflect on improvements in broadening the discussion about the practical implications of the findings for public health policy and what educational programs could be implemented in Saudi Arabia (e.g. awareness programs about peer influence as a predisposing factor for abuse problems; training programs for safer and more effective coping strategies to deal with the stressors of personal and academic life, assertiveness techniques...). It would also be beneficial to consider discussing the wider implications of the study's findings, potentially linking them to global trends and potential cultural factors that influence university students' knowledge and attitudes. The limitations of the study are acknowledged.
The references are comprehensive and relevant, corroborating the context and conclusions of the study. The inclusion of recent studies adds credibility to the research.

Overall, the article contributes relevant data to the field of university students' knowledge and attitudes about substance abuse, particularly in the Mecca region, Saudi Arabia. 

Author Response

Reviewer 3

The study addresses a topic of global relevance to public health, as it deals with the knowledge and attitudes of university students towards illicit substance abuse, which has serious consequences for physical, social and mental well-being.

  1. The abstract provides a clear summary of the context, methods, results and conclusions of the study. It is concise and covers the essential elements of the research.

We thank the reviewer for the positive feedback. We are pleased to know that the abstract clearly and concisely communicates the key aspects of the study.

  1. The introduction makes reference to the literature, the objectives of the study are clearly defined. However, a more explicit statement of the importance of the study and how it contributes to existing knowledge would be useful. In addition, it may also be useful to expand on the global context of knowledge and attitudes related to substance abuse among university students, namely statistical data, and how these trends may be relevant to Saudi Arabia.

We thank the reviewer for this constructive suggestion. In response, we have revised the Introduction to include a clearer statement on the importance of the study.

  1. The study methodology has several strengths, including a well-described sample size and study design. In addition, ethical considerations, including approval and consent procedures, are adequately addressed. Although the period of the study is mentioned, it would be beneficial to discuss why this period was chosen and how it may impact knowledge and attitudes about substance abuse among university students.

We thank the reviewer for this helpful suggestion. The study was conducted during the academic term, a period when students were actively engaged in their coursework.

  1. The results of the study were presented clearly, with detailed demographic information, easy-to-read and interpret tables and statistical tests appropriate to the aims of the study. However, it may be useful to organize the results section more cohesively, possibly using subheadings to improve readability.

We thank the reviewer for the positive feedback on the clarity and statistical presentation of the results. In response to the suggestion, we have reorganized the Results section using descriptive subheadings to improve cohesion and readability.

  1. The discussion section effectively relates the results of the study to the literature, referencing other studies on knowledge and attitudes about substance abuse in university students. However, they may reflect on improvements in broadening the discussion about the practical implications of the findings for public health policy and what educational programs could be implemented in Saudi Arabia (e.g. awareness programs about peer influence as a predisposing factor for abuse problems; training programs for safer and more effective coping strategies to deal with the stressors of personal and academic life, assertiveness techniques...). It would also be beneficial to consider discussing the wider implications of the study's findings, potentially linking them to global trends and potential cultural factors that influence university students' knowledge and attitudes. The limitations of the study are acknowledged.

We thank the reviewer for the insightful suggestions. The Discussion section has been expanded to address the practical implications of our findings, including relevant educational interventions in Saudi Arabia.

  1. The references are comprehensive and relevant, corroborating the context and conclusions of the study. The inclusion of recent studies adds credibility to the research.

We thank the reviewer for this positive feedback.

  1. Overall, the article contributes relevant data to the field of university students' knowledge and attitudes about substance abuse, particularly in the Mecca region, Saudi Arabia.

We sincerely thank the reviewer for this encouraging feedback.

Round 2

Reviewer 2 Report

Comments and Suggestions for Authors

The authors significantly improved the manuscript and included all necessary corrections in the Introduction, Methods and Discussion sections. 

Just one small editing error is present - in the Methods section, after the new paragraph was added, the previous information remained, the last sentence of section 2.1. should be deleted as it describes the power calculation.

Author Response

Thank you, we have deleted the repeated part.

Reviewer 3 Report

Comments and Suggestions for Authors

Parabéns pelas mudanças. Não tenho mais sugestões de melhorias no momento.

Author Response

Thank you for your valuable comments that improved the manuscript significantly.